# The Use of Cyclin-Dependent Kinase 4/6 Inhibitors in Elderly Breast Cancer Patients: What Do We Know?

**DOI:** 10.3390/cancers16101838

**Published:** 2024-05-11

**Authors:** Alexandre Giraudo, Renaud Sabatier, Frederique Rousseau, Alexandre De Nonneville, Anthony Gonçalves, Maud Cecile, Cecile Braticevic, Frederic Viret, Lorene Seguin, Maria Kfoury, Dorothée Naudet, Marie Hamon, Louis Tassy

**Affiliations:** 1Institute Paoli-Calmette, 13009 Marseille, France; sabatierr@ipc.unicancer.fr (R.S.); rousseauf@ipc.unicancer.fr (F.R.); alexandredenonneville@gmail.com (A.D.N.); goncalvesa@ipc.unicancer.fr (A.G.); cecilem@ipc.unicancer.fr (M.C.); bannierc@ipc.unicancer.fr (C.B.); viretf@ipc.unicancer.fr (F.V.); seguinl@ipc.unicancer.fr (L.S.); kfourym@ipc.unicancer.fr (M.K.); naudetd@ipc.unicancer.fr (D.N.); 2Medical School Department, Aix-Marseille Université, 13005 Marseille, France; marie.hamon@ap-hm.fr

**Keywords:** breast cancer, elderly patients, cyclin-dependent kinase inhibitors, CDK4/6, endocrine therapy, quality of life, dose adaptation, advanced breast cancer, adjuvant

## Abstract

**Simple Summary:**

This position paper aims to address specific clinical questions regarding the use of cyclin-dependent kinase 4/6 inhibitors in elderly patients with early or advanced breast cancer. Its objectives are to delineate the current state of knowledge regarding the efficacy of these treatments in the elderly population and their tolerance profile, including the impact on quality of life, with a particular focus on the frailest subgroups, and to attempt to define the optimal treatment strategy for elderly and fragile patients (dosage and therapeutic sequence).

**Abstract:**

Background: Breast cancer (BC) incidence increases with age, particularly in HR-positive/HER2-negative subtypes. Cyclin-dependent kinase 4 and 6 inhibitors (CDK 4/6is) alongside endocrine therapy (ET) have emerged as promising treatments for HR-positive/HER2-negative advanced and early BC. However, their efficacy, safety, and impact on quality of life (QoL) in older and frail patients remain underexplored. Methods: This position paper assesses the existing literature from 2015 to 2024, focusing on CDK4/6is use in patients aged 65 years and older with HR-positive/HER2-negative BC. Results: Our analysis methodically addresses critical questions regarding the utilization of CDK4/6is in the elderly BC patient population, organizing findings from the metastatic and adjuvant settings. In the metastatic setting, CDK4/6is significantly improve progression-free survival (PFS), paralleling benefits observed in younger patients, and suggest potential overall survival (OS) benefits, warranting further investigation. Despite an increased incidence of grade ≥ 3 adverse events (AEs), such as neutropenia and asthenia, CDK4/6is present a markedly lower toxicity profile compared to traditional chemotherapy, with manageable side effects. QoL analysis indicates that integrating CDK4/6is into treatment regimens does not significantly impact elderly BC patients’ daily life and symptom management. Special attention is given to frail subgroups, and personalized approaches are recommended to balance efficacy and adverse effects, such as starting with ET alone and introducing CDK4/6is upon progression in patients with a low disease burden. Transitioning to the adjuvant setting, early results, particularly with abemaciclib, indicate positive effects on disease-free survival (DFS), emphasizing the need for continued analysis to validate these findings and assess long-term implications. However, data on older patients are insufficient to conclude whether they truly benefit from this treatment. Conclusion: Overall, CDK4/6is present a favorable benefit-risk profile in older BC patients, at least in advanced BC; however, further research is warranted to optimize treatment strategies and improve outcomes in this population

## 1. Introduction

Breast cancer (BC) is significantly associated with aging [1,2], with the HR-positive/HER2-negative subtype predominating in patients older than 75 years [3]. The standard treatment in HR-positive/HER2-negative metastatic setting has evolved to include cyclin-dependent kinase 4 and 6 inhibitors (CDK 4/6is) alongside endocrine therapy (ET), showing promises in improving patient outcomes [4,5]. The introduction of CDK 4/6is in the early 2010s marked a significant advancement, with palbociclib leading the way [6], followed by abemaciclib and ribociclib. Overall survival (OS) analyses found a significant advantage for ribociclib plus letrozole over letrozole alone [7], as well as for abemaciclib associated with fulvestrant [8]. With palbociclib, a numerical trend toward OS improvement was also observed but did not reach statistical significance either when combined with letrozole or fulvestrant [9,10,11]. Furthermore, abemaciclib and ribociclib also found their place in the adjuvant setting following the results of MonarchE [12] and NATALEE trials [13].

Elderly BC patients, particularly those over 75 years, often receive less aggressive treatment and exhibit lower survival rates [14]. While healthier patients are treated similarly to younger ones, frail individuals may require different considerations. Their underrepresentation in clinical trials complicates the extrapolation of results to this population. For example, in PALOMA studies, patients aged 64 to 75 years constituted only 37% of study participants, and those over 75 years represent merely 9% of the population [6,15,16]. Likewise, in MONARCH 2 and 3 studies, less than 13% of all patients were over 75 years [17,18]; for the MONALEESA trials [19,20], the percentage of patients over 75 years old is not reported.

In an aging population with increasing numbers of patients aged 85 or above, this issue becomes particularly problematic. Nonetheless, CDK4/6is have been approved in Europe for ABC without age restrictions. The question then becomes: can CDK4/6is be prescribed in the same manner for older patients as for younger ones? How should we approach treatment for the oldest and frailest patients, for whom less data are available? We aim to explore these questions, focusing on patients with metastatic/advanced and unresectable tumors, through the following topics:A.EfficacyB.ToleranceC.Specific safety considerations regarding the frailest subgroupD.Impact on quality of lifeE.Key data points concerning dose adjustmentsF.Rational arguments regarding the choice of CDK4/6isG.Determining the optimal therapeutic sequence

Then, we address the question of their adjuvant use in patients with early breast cancer through the following areas of focus:A.EfficacyB.ToleranceC.Impact on quality of life

One should note that, although the definition of elderly patients is not consensual, most trials defined their elderly subgroup as patients aged 65 years and above, even though this population is highly heterogeneous. For consistency with these studies and the age-specific analyses derived from them, we have chosen this threshold to consider a patient as elderly. However, specific considerations will be discussed regarding the most vulnerable subgroups, i.e., very old patients (over 80 or even 85 years old) and frail patients due to their multiple comorbidities or existing limitations in autonomy.

## 2. Materials and Methods

This position paper relies on a comprehensive examination of literature from 2015 (publication date of the first pivotal trial of a CDK 4/6i) to 2024.

We conducted searches on PubMed using keywords related to “Cyclin-Dependent Kinase 4/6 Inhibitors”, “palbociclib or ribociclib or abemaciclib”, “Breast Cancer”, “Elderly or older”, “Aging”, “Safety”, and “Efficacy” in various combinations and selected studies analyzing CDK4/6is use in patients aged 65 years and older with HR-positive/HER2-negative BC, encompassing randomized controlled trials (RCTs) and real-world studies (RWSs).

Given that our study is not properly speaking a systematic review, we did not adhere to the PRISMA guidelines.

## 3. Results

### 3.1. Patients with Advanced Breast Cancer (Metastatic or Unresecable)

#### 3.1.1. Efficacy

Regarding efficacy, a FDA pooled data analysis from three pivotal trials (PALOMA 2, MONALEESA 2, and MONARCH 3) emphasized the effectiveness and safety of CDK 4/6is in women over 70 and 75 years old [21]. In patients over 70, the hazard ratio (HR) for progression-free survival (PFS) of the combination therapy compared to aromatase inhibitors (AI) alone was 0.52 (95% CI [0.38–0.70]) with a median PFS of 33.1 months (95% CI, [27.8-not evaluable]) versus 19.2 months (95% CI, [14.7–26.0]), respectively (Table 1). These outcomes closely mirrored those of younger patients, with a HR of 0.57 (95% CI, [0.48–0.65]) and a median PFS of 27.3 months (95% CI, [23.1–27.7]) versus 14.1 months (95% CI, [12.9–15.9]) with AI alone. In individuals aged 75 and older, the HR for PFS was 0.49 (95% CI [0.31–0.76]), favoring the combination therapy, with an estimated median PFS of 31.1 months (95% CI, [20.2–not reached]) versus 13.7 months (95% CI, [10.9–24.9]) with AI alone.

Additionally, a meta-analysis from phase 2 and 3 clinical trials focusing on the outcomes of patients 65 years and older receiving CDK4/6is with ET showed a PFS improvement compared to ET alone with a HR of 0.77 (95% CI, [0.62–0.95]) [22]. Another pooled analysis focusing only on the MONARCH program did not identify a statistically significant difference in the PFS improvement with abemaciclib compared to placebo across three selected age groups (<65, 65 to 74, and >75) [23].

Moreover, a recent retrospective German RWS on the use of the three CDK4/6is across a broad population supported these findings, showing that PFS benefits are independent of age [24].

Concerning OS, a meta-analysis of the pivotal trials of the three compounds demonstrated that adding a CDK4/6i to ET offers benefits regardless of age (±65 years) [25]. It is also worth mentioning an American retrospective RWS compared first-line palbociclib plus AI versus AI alone in a cohort of 2888 patients with a median age of 70 years. After adjusting based on propensity score matching, they reported a median OS of 57.8 months (95% CI, [47.2–not estimable]) in the palbociclib group compared to 43.5 months (95% CI, [37.6–48.9]) in the AI group, with a HR of 0.72 (95% CI, [0.62 to 0.83]; *p* < 0.0001) [26].

In summary, both RCTs and the RWS demonstrate that elderly patients benefit from CDK4/6is in terms of PFS, akin to their younger counterparts. We still need further explorations to assess the CDK4/6is impact on OS among older patients.

**Table 1 cancers-16-01838-t001:** Progression-free survival (PFS) and overall survival (OS) outcomes in the whole population and the ≥65 years old (yo) population in princeps trials with palbociclib (PALOMA program), ribociclib (MONALEESA program) and abemaciclib (MONARCH program) [6,7,8,9,10,15,16,17,18,19,20,27]. Two real-world studies (RWSs) were also included [24,26]. The median durations are given in months with the hazard ratio (HR) and 95% confidence interval when available.

	Study Name and Molecule	Study Design	Total Effective	% ≥ 65 yo Included	PFS (Whole Population)	OS (Whole Population)	PFS (>65 yo)	OS (>65 yo)
**First line only**	PALOMA 1 *palbociclib + letr* *ozole*	Phase II RCT	165	46	20.2 vs. 10.2 HR 0.488 *p* = 0.0004	37.5 vs. 34.5 HR 0.897 *p* = 0.281 (NS)	HR 0.505 (0.269–0.948)	HR 0.97 (0.57–1.67)
	PALOMA 2 *palbociclib + letrozole*	Phase III RCT	666	40.8	27.6 vs. 14.5 HR 0.563 *p* < 0.001	NA	HR 0.57 (0.388; 0.837)	NA
	MONALEESA-2 *ribociclib + letrozole*	Phase III RCT	668	44.1	25.3 vs. 16 *p* = 9.63 × 10⁻⁸	63.9 vs. 51.3 *p* = 0.008	HR 0.610 (0.393–0.947)	HR 0.87 (0.64–1.17)
	MONARCH 3 *abemaciclib + any AI*	Phase III RCT	493	45	NR vs. 14.7 HR 0.54 *p* = 0.000021	66.8 vs. 53.7 HR 0.804 (0.637–1.015) *p* = 0.064	HR 0.57 (0.36–0.90)	NA
	American real-world study [26] *palbociclib + any AI*	Retrospective comparative RWS with propensity score matching	2888	68	19.8 vs. 14.9 HR 0.72 [0.63–0.82] *p* < 0.0001	57.8 vs. 43.5 HR 0.72 [0.62–0.83] *p* < 0.0001	NA	HR * 0.72 [0.57–0.90] HR ** 0.69 [0.52; 0.91]
**Second line** **Only**	PALOMA-3 *palbociclib + fulvestrant*	Phase III RCT	521	24.8	11.2 vs. 4.6 HR 0.49 *p* < 0.00001	34.9 vs. 28 HR 0.81 *p* = 0.09	NA	NA
	MONARCH-2 *abemaciclib + fulvestrant*	Phase III RCT	669	36.6	16.4 vs. 9.3 HR 0.553 *p* < 0.001	46.7 vs. 37.3 *p* = 0.01 HR 0.757	HR 0.620 (0.447–0.860)	0.898 (0.638–1.263)
**First or second line**	MONALEESA-3 *ribociclib + fulvestrant*	Phase III RCT	726	46.6	20.5 vs. 12.8 *p* < 0.001	53.7 vs. 41.5 HR 0.726 *p* = 0.004	HR 0.597 (0.436–0.818)	HR 0.72 (0.53–0.99)
**Any line**	German RWS [24] *Palbociclib, ribociclib, or abemaciclib + any ET*	Retrospective noncomparative RWS	448	NA (median age: 63)	17	NA *(independent of age)*	NA	NA

* HR in the 65–74 yo subgroup; ** HR in the ≥75 yo subgroup.

#### 3.1.2. Tolerance

The main CDK 4/6i adverse events (AEs) are hematologic (primarily neutropenia), gastrointestinal (principally diarrhea), biological (notably transaminase elevation), and nonspecific adverse effects, such as fatigue and reduced appetite.

Palbociclib and ribociclib exhibit less specificity in inhibiting CDK4 compared to abemaciclib, correlating with heightened incidences of neutropenia. Furthermore, ribociclib has been implicated in QT interval prolongation, whereas abemaciclib demonstrates a greater prevalence of diarrhea than its counterparts [28].

These agents however remain considerably less toxic than chemotherapy, as evidenced in the PEARL trial, which indicated more toxicity and accelerated QoL deterioration in patients treated with capecitabine compared to a CDK4/6i [29]. The incidence of grade ≥ 3 neutropenia was tenfold higher with palbociclib than with capecitabine (55% vs. 5.5%), but the rate of febrile neutropenia was similar (about 1%).

Similarly, a meta-analysis of pivotal trials focusing on hematological AEs found a significant increase in grade ≥ 3 leukopenia but not febrile neutropenia [30].

Regarding older patients (Table 2), the FDA pooled analysis highlighted a higher incidence of grade ≥ 3 AEs (88.8% over 75 years old versus 73.4% before 75 years old) [21]. Similarly, the previously cited pivotal trial meta-analysis reported that elderly (≥65 years old) patients receiving palbociclib showed higher susceptibility to neutropenia, anemia, back pain, asthenia, and infections than their younger counterparts. With ribociclib, they also were more likely to experience neutropenia (OR 2.7, 95% CI [1.3; 5.4]) and hypertension (OR 85, 95% CI [15; 475]) [22]. This aligns with an investigation of the Spanish subset of the COMPLEEMENT trial, where grade ≥ 3 neutropenia was observed in 59.5% of the total cohort and 77% of participants aged over 70 years [31].

With abemaciclib, grade ≥2 diarrhea was observed more frequently among older women (<65 years old: 39.5%; 65–74 years old: 45.2%; ≥75 years old: 55.4%) in the age-specific MONARCH 2 and 3 subgroup analysis [23].

Finally, regardless of the CDK4/6i, asthenia is reported in approximately 40% of the FDA pooled analysis patients, increasing to over 50% in ≥70 years old [21].

In summary, CDK4/6is exhibit higher toxicity in older patients compared to their younger counterparts but remain significantly less harmful than conventional chemotherapy. The most common AE, neutropenia, appears less clinically significant with CDK4/6is than with cytotoxic chemotherapy given the low incidence of infectious complications. However, the impact of asthenia in older patients, which may lead to reduced activity and frailty, should not be underestimated.

#### 3.1.3. Specific Safety Considerations Regarding the Frailest Subgroup

Among elderly patients, vulnerability due to comorbidities is common. In addition, specific considerations may be needed for the oldest subgroups (over 85 years old for example). While some patients may benefit from CDK4/6is with minimal AEs, the most vulnerable may require a tailored approach prioritizing QoL over survival extension.

Currently, there are no definitive guidelines for prescribing CDK 4/6is in this subgroup. Even though most toxicities appear manageable, they may precipitate geriatric decompensation in frail individuals. Specific concerns include the following:Neutropenia: beyond the hypothetical risk of infections, it may demand more intensive monitoring, including regular blood tests, and cause anxiety and discomfort.Diarrhea: even mild to moderate cases can significantly affect these patients, potentially leading to dehydration and renal dysfunction.Asthenia, which may further impair or lead to a loss of independence.Loss of appetite, which is particularly problematic in a population at risk of undernutrition and sarcopenia.The risk of falls, which can be exacerbated by some of the aforementioned complications (dehydration, sarcopenia, loss of autonomy in activities of daily living), can quickly lead to a cascade of geriatric decompensation.

The findings from PALOMAGE, a French prospective study assessing palbociclib and ET in real-world settings for women aged ≥70 years were recently published [32]. Among the 807 participants (median age 79), 68.3% scored ≤ 14 on the G8, indicating frailty, and 17.9% had an ECOG score ≥ 2. A total of 70% encountered at least one AE, with 43.1% experiencing grade 3–4 AEs.

Neutropenia was the most common AE, occurring in 43% of patients, with a febrile neutropenia rate comparable to other study populations (1%). Additional AEs included asthenia (16%), anemia (17%), and thrombocytopenia (13%). While frailty factors did not significantly affect the occurrence of grade ≥ 3 toxicities, AEs were more common in heavily pre-treated patients.Haut du formulaire

Interestingly, the study showed that palbociclib did not worsen monitored geriatric parameters at 3 and 6 months. However, frailty was associated with a greater chance of permanent treatment discontinuation at 6 months.

In summary, the absence of geriatric-specific data in pivotal trials poses a risk when applying the findings to the frailest groups. However, real-world evidence (particularly concerning palbociclib) supports the use of CDK4/6is in these populations. Given the minimal risk of febrile neutropenia, we should not increase the frequency of blood test monitoring beyond what is customary for younger patients.

#### 3.1.4. Impact on Quality of Life

Patient preference remains a crucial consideration given that older women might prioritize maintaining QoL over extending survival. Consequently, assessing AEs and patient-reported outcomes (PROs) is as essential as evaluating efficacy.

Reports from pivotal trials have included PROs. Across the entire cohort of phase 3 studies, there was no significant reduction in QoL scores during treatment with a CDK4/6i with ET compared to ET alone. Notably, an improvement in QoL and better pain management with palbociclib combined with fulvestrant was observed in PALOMA 3 (in patients who had already progressed on ET [34]. Similarly, employing ribociclib or abemaciclib with ET resulted in enhanced pain control in MONALEESA 2 and MONARCH 2, respectively. None of the MONALEESA and MONARCH trials reported a significant overall decline in QoL [35].

Concerning older patients, the real-world data from PALOMAGE are available, in which any QoL deterioration with palbociclib was reported in its predominantly frail population. A clinically meaningful decrease in pain was recorded at 3 and 6 months compared to baseline. However, the study’s design does not allow conclusions that this improvement resulted from adding palbociclib to standard ET rather than from ET alone.

In summary, there is no evidence that adding a CDK 4/6i to ET alters QoL in older or frail patients. However, as available data remain too limited to draw any firm conclusions, we suggest a cautious approach regarding QoL with CDK4/6is, especially in asymptomatic patients.

#### 3.1.5. Key Data Points Concerning Dose Adjustment

Dose reduction protocols for CDK4/6is in ABC already exist. Palbociclib’s starting dose of 125 mg daily can be decreased to 100 mg and further to 75 mg as needed. Abemaciclib begins at 150 mg twice daily, with possible reductions to 100 mg and then to 50 mg. For ribociclib, the initial dose is 600 mg daily, reducible to 400 mg or 200 mg daily.

There are some data regarding unplanned dose reductions (due to AEs) and the impact of received dose intensity on efficacy. A study examining factors affecting PFS in the PALOMA 2 cohort showed no significant difference between patients who underwent palbociclib dose reduction due to AEs (at 100 mg or 75 mg daily) and those who did not [36].

Similarly, dose reductions of ribociclib due to AEs in the MONALEESA trial series did not result in a decrease in PFS or response rates among patients with a relative dose intensity between 72% and 96% or inferior to 72%. Notably, 45% of patients required dose adjustments due to AEs [37].

In the German retrospective RWS, 29% of patients needed dose reductions (19% for hematologic AEs). Interestingly, patients requiring dose adjustments had a slightly better PFS compared to those who did not. The authors suggested that dose reduction may reduce therapy interruptions, ensuring steady drug levels and potentially improving efficacy [24]. These findings were consistent with two other retrospective RWSs focusing on palbociclib [33,38].

Regarding the temptation to initiate treatment at a lower dose in elderly patients, a small retrospective American RWS indicated that this approach appeared to result in lower response rates and shortened PFS with palbociclib [39].

In summary, there is no evidence supporting the initiation of treatment at a reduced dose. However, close monitoring, particularly in frail patients, is essential. Prompt dose adjustment should be considered if toxicity becomes unmanageable, as it does not appear to compromise efficacy.

#### 3.1.6. Rational Arguments Regarding the Choice of CDK4/6i

Concerning efficacy, none of the three molecules has proven superior as they have not been directly compared in frontline settings. Although OS data (from pivotal trials secondary endpoints) slightly favor ribociclib (OS improvement with letrozole and fulvestrant) over abemaciclib (OS improvement only with fulvestrant) and palbociclib (no OS improvement), such statistics scarcely assist clinicians in selecting one drug over another.

Currently, the decision remains influenced by toxicity profile and physician’s and patient’s preference. For instance, prescribing abemaciclib to a patient with chronic diarrhea or palbociclib to a patient with a myelodysplastic syndrome could be challenging. Additionally, ribociclib might be contraindicated in patients on QT-prolonging medications.

In other words, while ribociclib or abemaciclib may be preferred over palbociclib due to more robust OS data in ABC, ribociclib may be favored in frail populations due to diarrhea concerns. If ribociclib and abemaciclib are contraindicated or challenging to introduce because of comorbidities or comedications, palbociclib remains a reasonable choice, prioritizing tolerability over marginal efficacy.

#### 3.1.7. Determining the Optimal Therapeutic Sequence

Therapeutic strategy delineation remains complex, as highlighted by a 2022 publication indicating a lack of consensus among practitioners regarding optimal approaches [40].

The 2023 ESMO guidelines reaffirm a CDK4/6i with ET as standard regardless of age, though with an elevated risk of hematologic AEs. ET alone is recommended for patients with poor ECOG status or significant comorbidities (although not identified through specific clinical or biological markers) [5].

Similarly, the Young International Society of Geriatric Oncology suggests starting with ET alone for frail elderly patients with minimal symptoms and low tumor burden and considering CDK4/6is for those with threatening or symptomatic metastases and upon disease progression [41].

SONIA, a RCT comparing the use of CDK4/6is with ET in a first-line setting versus their introduction in a second-line setting after progression on ET alone might however change the game [42]. The preliminary results presented at the 2023 ASCO meeting did not demonstrate a significant benefit for using a CDK4/6i as a first-line treatment. The median PFS2 (time from randomization to second disease progression or death) was 31.0 months with the first-line strategy and 26.8 months with the second-line strategy (HR 0.87; 95% CI: 0.74 to 1.03; *p* = 0.10) [43].

Additionally, the first-line strategy resulted in longer CDK4/6i exposure (median 24.6 vs. 8.08 months), more adverse events (70% more grade ≥ 3 events), and higher costs. While this study did not specifically focus on elderly subjects, these findings support starting with ET alone in frail patients.

However, this trial has limitations. With a median follow-up duration of only 37.3 months, OS data are lacking to reach a definitive conclusion. Additionally, 90% of patients received palbociclib, which limits the applicability of its findings to ribociclib or abemaciclib.

In summary, present data support combining CDK4/6is with ET in first-line settings regardless of age. However, trials like SONIA may prompt adjustments if the introduction of CDK4/6is in first-line setting results in no OS advantage compared to second-line use of CDK4/6is.

Currently, for symptomatic patients or those with high tumor burden, first-line ET combined with CDK 4/6i remains advised. However, in frail and/or minimally symptomatic patients, monotherapy ET as a first line and CDK4/6i addition upon progression seems to be a reasonable option.

### 3.2. Patients with Early Breast Cancer: Adjuvant Setting

#### 3.2.1. Efficacy

Currently, abemaciclib is the sole CDK4/6i endorsed for use in an adjuvant context. Grounded on findings from the MonarchE trial [12], abemaciclib is indicated alongside standard ET for two years at a dosage of 150 mg twice daily if the patient fulfills either of the following criteria:At least 4 axillary nodes invaded.From 1 to 3 axillary nodes invaded and one of the following criteria: grade 3 or primary tumor size > 5 cm.

Within this population, a relative risk reduction of 25% for invasive disease recurrence was observed at a two-year follow-up, and extended data up to four years show sustained benefit. The disease-free survival (DFS) rates were 92.2% for the abemaciclib cohort versus 88.7% for the ET alone group at two years and 85.8% versus 79.4% at four years, respectively [44].

Regarding OS, no significant improvement was found after a five-years follow-up, despite fewer deaths in abemaciclib-treated patients. Given the delayed recurrence nature of HR-positive BC and advancements in metastatic patient survival rates, longer follow-up is required to assess OS impact [45].

However, only 15% of participants were aged 65 or older, with a median age of 51 years, which is approximately a decade younger than those in pivotal studies of CDK 4/6is in ABC. Subgroup analysis from a 42-month follow-up suggested older patients (>65 years) might benefit from this treatment, although not statistically significant (HR = 0.767, 95% CI: 0.556, 1.059), likely due to the small sample size [46].

Regarding ribociclib, the results of the NATALEE trial were recently published. Contrary to abemaciclib in MonarchE, ribociclib was given at a lower dose than the metastatic standard (400 mg daily versus 600 mg) over a three-year period. The trial’s inclusion criteria were also wider, allowing node-negative patients with additional risk factors like grade 3 disease, grade 2 and Ki-67 ≥ 20%, or a high-risk genomic signature [47].

They found a significant increase in DFS with ribociclib and ET compared to ET alone (HR 0.75; 95% CI 0.62–0.90; *p* = 0.0014), showcasing a three-year DFS rate of 90.4% with ribociclib and ET versus 87.1% with ET alone.

Yet, the median age of this trial’s population was 52 years, and no specific age-related analysis is available [48].

For palbociclib, two trials have evaluated its adjuvant efficacy. The PALLAS trial explored the addition of palbociclib to adjuvant ET during the first two years for early-stage BC patients (stage 2 or 3) [49], and the PENELOPE-B trial assessed the addition of palbociclib for one year in high-risk patients with residual disease post-neoadjuvant treatment [50]. Both trials are negative for primary outcomes.

#### 3.2.2. Tolerance

In the MonarchE trial, the tolerability of abemaciclib paralleled that noted in metastatic patients. Dose modifications due to AEs concerned 68% of participants, though only 16% definitively ceased treatment.

The age-specific safety analysis highlighted that older patients were more likely to encounter grade ≥ 3 AEs (67% for those aged ≥ 75 years versus 49% for those ≤65 years). Nevertheless, caution is warranted in interpreting these findings due to the small elderly cohort within the trial population and absent geriatric data.

Regarding the NATALEE trial, its safety profile was reported as favorable, yet age-specific data have not been disclosed.

#### 3.2.3. Impact on Quality of Life

For patients without visible remaining disease, there is not the same chance to improve QoL by dealing with disease-related symptoms as in the metastatic context, at least during the treatment period. The PROs from the MonarchE trial underscore the tolerability of abemaciclib combined with ET in early BC, showing comparable patient-reported health-related QoL and patient experiences of being ‘bothered by side-effects of treatment’ relative to ET alone. However, due to the small number of elderly and frail patients in MonarchE, firm conclusions cannot be drawn for this population [51].

In summary, abemaciclib and ribociclib improve DFS in the adjuvant setting, with benefits extending to the elderly population, notably for abemaciclib. Yet, no OS benefit was established. Although young patients might gain significant OS improvement, such outcomes are unlikely in older and frail individuals, due to their reduced life expectancy and higher risk of dying from non-cancer causes.

Thus, based on current evidence, we propose the following recommendations regarding the prescription of abemaciclib in the adjuvant setting for elderly patients:Abemaciclib should be avoided for those who are frail and/or have a short life expectancy because of significant comorbidities.Abemaciclib should be considered on an individual basis for those who are healthy with a relatively long-life expectancy.

Should the adjuvant use of ribociclib be approved, its favorable tolerance profile may allow broader prescription among the elderly.

## 4. Discussion

CDK4/6is show efficacy in the older population, improving PFS and potentially OS, especially in metastatic cases, with manageable side effects and a favorable benefit–risk ratio even in the frailest and oldest subgroups.

However, the lack of data in this population underlies the need for more inclusive clinical trials and RWSs with detailed geriatric assessment. Such studies should also manage to collect PROs, even among the very elderly. Ideally, all the geriatric supportive care aiming to ensure the preservation of autonomy and QoL (such as nutritional and social follow-up, nursing measures, and kinesiotherapy) should also be monitored in future studies, which would entail additional costs.

As clinicians aim to balance treatment efficacy with QoL for older and frail ABC patients, factors influencing treatment selection include the following:The benefits of PFS: delaying the first progression and the apparition/worsening of symptoms can be a goal even without OS prolongation.The toxicity profiles of each medication.Patient preferences, necessities, and comorbidities.

Subgroup analyses help tailor treatments to individual needs, with some patients benefiting from less toxic options such as ET alone, particularly those with a low disease burden, advanced age, and a preference for maintaining independence. This strategy aligns with current ESMO guidelines and is supported by initial findings from SONIA. If such trials confirm an absence of OS benefit with first-line versus second-line CDK4/6i use, this approach might emerge as a new norm for most patients.

## 5. Conclusions

Therefore, for the frailest ABC patients with a low disease burden and few tumor symptoms, we suggest starting with ET alone and introducing a CDK4/6i upon progression.

Among the three CDK4/6is available, ribociclib seems to have the best benefit–risk ration for older ABC patients.

When employed, CDK4/6is should be initiated at standard dosing, with a close monitoring and consideration for early dose reduction upon the onset of toxicities.

Regarding the adjuvant setting, forgoing CDK4/6i use in old or frail patients appears reasonable given the lack of data for this population. The decision to prescribe them should be made on a case-by-case basis, recognizing that those with the longest life expectancy stand to benefit the most from this therapy.

## Figures and Tables

**Table 2 cancers-16-01838-t002:** Rate of selected CDK4/6is adverse events in age-specific subgroups from clinical trials and real-world studies. The numbers represent toxicity occurrence rates in % of the subgroup population. yo: years old. [21,23,31,32,33].

	Clinical Trials								Real-World Studies	
	All CDK4/6is			Ribociclib		Abemaciclib			Palbociclib	
	FDA Pooled Analysis			COMPLEEMENT Trial (Spanish Population)		MONARCH 2–3 Age-Specific Subgroup Analysis			PALOMAGE	English RWS [33]
	<65 yo	≥65 yo	≥75 yo	Whole Population	≥70 yo	<65 yo	65–74 yo	≥75 yo	≥70 yo	≥75 yo
Diarrhea (any grade)	43	51	53.6	15.8	16	85	83.6	85.5	4.1	18.5
Grade ≥ 3 diarrhea	2.9	4.8	7.2	0.8	0	9.9	12.8	19.3	0.4	1.1
Asthenia (any grade)	42.5	49.1	54.4	37.8	41.3	34.8	48.4	51.8	16.3	53.6
Grade ≥ 3 neutropenia	51.8	53.9	53.6	59.5	64	25.8	27.4	18.1	32.4	46.4
Grade ≥ 3 hepatotoxicity	6.7	6.3	6.4	NA	NA	2.8	3.2	2.4	NA	1.3

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
