# Peer review of "The Use of Cyclin-Dependent Kinase 4/6 Inhibitors in Elderly Breast Cancer Patients: What Do We Know?"

_cancers, 2024, doi:10.3390/cancers16101838_

Round 1

Reviewer 1 Report

Comments and Suggestions for Authors

This is a review article/position paper related to the use of cyclin-dependent kinase 4/6 inhibitors for the treatment of elderly breast cancer patients. The article focuses on patients age 65 or older with estrogen receptor positive and HER2 negative breast cancer. They find that like in younger patients, elderly patients with metastatic disease benefit from CDK4/6i for extending progression-free survival. There are known adverse events including neutropenia which were also observed in these patients; frail patients are recommend to start with endocrine therapy alone and continue to CDK4/6i therapy when progression arises. As a monoagent, abemaciclib was found to provide positive benefits.  Overall, this is a well written paper, that address an appropriate set of questions that focus on the use of these compounds in metastatic patients. The authors address each questions with an appropriate set of data and publications.

Minor comment. The authors should consider using the same font for the tables. 

Author Response

Dear reviewer, first and foremost, I would like to express my gratitude for the time you have dedicated to reviewing our work. Thank you for your kind and constructive feedback.

Indeed, the font was not consistent across the two tables. We did not notice this during proofreading. Thank you for bringing it to our attention.

Reviewer 2 Report

Comments and Suggestions for Authors

In general, the manuscript gives a good overview of the current literature, focusing on elderly patients with indication for CDK 4/6i.

I do have some critics concerning the style of the manuscript.

1.    You give two sections on the metastatic setting (3.1.) and adjuvant setting (3.2.), but the headings should be improved:

Line 107: «This Metastatic (or unresectable tumor) patients”

There is no “unresectable tumor patient”, but patients with metastatic disease or unresectable tumor. Please correct.

Line 321 “Adjuvant setting”

Very short, the patients are missing in the title.

2.    I would not give subtitles with a question in the results section. (e.g., “Are CDK 4/6i as effective in older patients as in young patients with ABC?”), but rather make a statement.

Comments on the Quality of English Language

see above.

Author Response

Dear reviewer, first and foremost, I would like to express my gratitude for the time you have dedicated to reviewing our work. Thank you for your kind and constructive feedback.

I have edited, as you suggested, the titles of each subsection, which are no longer formulated as questions. Likewise, I have also reformulated the headings of the two sections using a patient-centered formulation.

Reviewer 3 Report

Comments and Suggestions for Authors

The topic is very interesting and well written.

You did not provide a clear definition of elderly patient

From what age do you consider a patient to be 'elderly'?

Author Response

Dear reviewer, first and foremost, I would like to express my gratitude for the time you have dedicated to reviewing our work. Thank you for your kind and constructive feedback.

Regarding the definition of elderly patients, we have chosen the threshold of 65 years for consistency with most princeps studies that use this threshold to define their elderly subgroup. You are correct in pointing out that this is not clearly stated in the text. I have added a paragraph at the end of the introduction to clarify this.